# Research on the Fault Diagnosis Method of an Internal Gear Pump Based on a Convolutional Auto-Encoder and PSO-LSSVM

**DOI:** 10.3390/s22249841

**Published:** 2022-12-14

**Authors:** Jian Liao, Jianbo Zheng, Zongbin Chen

**Affiliations:** 1Institute of Vibration and Noise, Naval University of Engineering, Wuhan 430033, China; 2Naval Key Laboratory of Ship Vibration and Noise, Naval University of Engineering, Wuhan 430033, China

**Keywords:** internal gear pumps, fault diagnosis, convolutional auto-encoder, multi-scale permutation entropy, least squares support vector machine

## Abstract

The raw signals produced by internal gear pumps are susceptible to noises brought on by mechanical vibrations and the surrounding environment, and the sample count collected during the various operating periods is not distributed evenly. Accurately diagnosing faults in internal gear pumps is significantly complicated by these factors. In light of these issues, accelerated life testing was performed in order to collect signals from an internal gear pump during various operating periods. Based on the architecture of a convolutional auto-encoder network, preprocessing of the signals in the various operating periods was performed to suppress noise and enhance operating period-representing features. Thereafter, variational mode decomposition was utilized to decompose the preprocessed signal into multiple intrinsic mode functions, and the multi-scale permutation entropy value was extracted for each intrinsic mode function to form a feature set. The feature set was subsequently divided into a training set and a test set, with the training set being trained to utilize a particle swarm optimization–least squares support vector machine network. For pattern recognition, the test set samples were fed into the trained model. The results demonstrated a 99.2% diagnostic accuracy. Compared to other methods of fault diagnosis, the proposed method is more effective and accurate.

## 1. Introduction

Internal gear pumps are characterized by low structural vibration and low outlet flow pulsation during operation, hence their application in marine hydraulic systems is on the rise. However, due to the harsh working environments encountered by marine hydraulic systems, the internal gear pump, which is the primary component of this type of system, undergoes rapid deterioration over the course of its lifetime, which can lead to potential safety hazards during long-term operation. If the internal gear pump fails, the marine hydraulic system will be severely compromised, if not destroyed [1]. Therefore, it is essential to conduct fault diagnostics and evaluate the internal gear pump operating condition. This should be accomplished such that replacement or maintenance recommendations can be made prior to the departure of the ship.

There are currently two issues associated with diagnosing faults in internal gear pumps. First, the operating state features must be transmitted to the vibration acceleration sensor on the casing through complex paths, such as rotor-oil casing, which causes the features to degrade and be easily obscured by mechanical vibration or environmental noise. Second, the number of samples collected during the life cycle of the internal gear pump is unbalanced, with far more samples collected during normal operating conditions than abnormal ones. Consequently, the model training method that employs a large number of sample data is not applicable to real-world applications [2]. Therefore, it is crucial to study the signal preprocessing method, feature extraction method, and pattern recognition method for the purpose of minimizing background noise, accurately extracting the fault features, and precisely determining the operational state of the internal gear pump.

Numerous signal preprocessing techniques, such as Fourier transformations, wavelet transformations (WTs) [3,4,5,6], and empirical mode decompositions (EMDs) [7,8,9,10], have been extensively proposed and implemented over the past few decades. In the majority of signal preprocessing strategies, a signal is divided into multiple frequency components or intrinsic mode functions, and the noise components are removed by establishing soft thresholds. However, traditional signal processing methods are incapable of accurately separating the fault features signal from complex vibrations and noises when applied to nonlinear and unstable industrial data.

Deep learning can automatically extract deep signal features from raw data, rendering it suitable for processing complex vibration data [11,12,13]. Specifically, convolutional auto-encoders (CAEs) have demonstrated outstanding image-denoising performance [14,15]. Yang et al. [16] proposed a residual wide-kernel deep convolutional auto-encoder for fault diagnosis of intelligent rotating machinery. In the study, a CAE was utilized to preprocess the raw signal of rotating machinery. The proposed method significantly enhanced the accuracy of fault diagnosis. AE Convolutional auto-encoders and convolutional neural networks trained by Prosvirin et al. [17] autonomously extracted global and local features from kurtograms. The feature vectors were propagated through artificial neural networks with a shallow structure to complete fault identification with positive results. Qian et al. [18] utilized a CAE as a signal feature extractor, and it produced an effective suppression of background noise. This fault diagnosis model exhibited high diagnostic precision and excellent generalizability. Unfortunately, the majority of deep learning models require a large quantity of labeled data for accurate recognition. Due to the unbalanced number of samples in different operating periods of internal gear pumps, the number of samples in fault states is frequently insufficient for deep learning models.

In recent nonlinear dynamic parameter analyses, the methods of measuring the uncertainty change in a signal by the entropy change, such as the approximate entropy, sample entropy, and fuzzy entropy methods, have been applied more frequently for identifying faults in rotating machinery. However, these methods are typically fragile and unable to fully describe the fault features [19,20,21]. In contrast, the multi-scale permutation entropy (MPE) method has excellent robustness for complex time-series, robust anti-interference, and simplistic calculations. In addition, the method is capable of analyzing signals at various time scales, yielding comprehensive data. Zheng et al. [22] used the MPE method to extract the fault features of rolling bearings, and then used an extreme learning machine to complete pattern recognition, a successful process.

In this study, a CAE was used to preprocess the vibration signal of an internal gear pump to address the low signal-to-noise ratio problem associated with raw signals. In light of the features of the original signal, a CAE network architecture was designed to suppress background noise and enhance the features of the operating state. In addition, variational mode decomposition (VMD) was utilized to decompose the signal into multiple intrinsic mode function (IMF) components in order to address the problem of a small number of samples in a fault state. The MPE values during various operating periods were extracted to form a feature set, which was then fed into a particle swarm optimization–least squares support vector machine (PSO–LSSVM) model for fault diagnosis of an internal gear pump.

## 2. Proposed Method

### 2.1. Convolutional Auto-Encoder

An auto-encoder (AE) is an unsupervised learning algorithm that learns data features by minimizing reconstruction errors. An AE network architecture typically consists of an input layer, a hidden layer, and an output layer. The input layer and the hidden layer constitute an encoder that produces a low-dimensional representation of the input data. The hidden layer and output layer constitute a decoder, which reconstructs the data from the hidden layer to produce input data with sample features.

Assuming that the input layer, x, is expressed by $\left\{x^{1},x^{2},\cdots ,x^{n}\right\}$, the *n*-dimensional vector can be mapped to the *m*-dimensional hidden layer vector, *h*, using the encoding function, f(x), to reduce the data dimension:(1)$$h=f(x)=s_{f}(W_{1}x+b_{1})$$

In Equation (1), $s_{f}$ represents the activation function of the encoder, $W_{1}$ is the weight matrix from the input layer to the hidden layer, and $b_{1}$ is the bias from the input layer to the hidden layer.

In the decoding stage, the decoding function, $s_{g}$, can be expressed by Equation (2):(2)$$y=g(h)=s_{g}(W_{2}h+b_{2})$$
where $s_{g}$ represents the activation function of the decoder, $W_{2}$ is the weight matrix from the hidden layer to the output layer, and $b_{2}$ is the bias from the hidden layer to the output layer. Finally, an error loss function was developed to evaluate the quality of training:(3)$$L_{AE}(x,y)=\frac{1}{2}{\left\|\left(x-y\right)\right\|}^{2}$$

A CAE is an auto-encoder that encodes and decodes using convolutional neural networks. It includes an input layer, several hidden layers, and an output layer. A CAE can extract signal features automatically while maintaining translation invariance. Leveraging the three-layer network in Figure 1 as an example, the encoder and decoder can be expressed by Equations (4) and (5):(4)$$x_{n}^{j}=f_{en}=S(x\cdot W_{1}^{i}+b_{1})$$
(5)$$\bar{x}=f_{de}=S(\sum_{i=1}^{N}x_{n}^{j}\cdot W_{2}+b_{2}^{i})$$

In Equations (4) and (5), $W_{1}^{i}$ denotes the weight of the *i*th convolution kernel in the encoder; $W_{2}$ indicates the weight of the convolution kernel in the decoder; and $b_{1}$ and $b_{2}^{i}$ represent the bias of the encoder and decoder, respectively. x, $x_{n}^{j}$, and $\bar{x}$ represent the raw data; the output of the *i*th convolution kernel in the encoder, and the reconstructed data, respectively. S is the nonlinear activation function, and N denotes the number of convolution kernels. The corresponding loss function is given by Equation (6):(6)$$L_{ae}={\sum_{i=1}^{n_{ac}}\left\|x-\bar{x}\right\|}^{2}$$
where $n_{ac}$ represents the number of training samples for the unsupervised learning.

### 2.2. Multi-Scale Permutation Entropy

The MPE calculations consisted of two distinct phases. First, multiscale coarse-graining was performed on the time series. The permutation entropy of coarse-grained sequences was then calculated at various scales.

(1) Multi-scale coarse-graining of a one-dimensional time series, $X=\left\{x_{i},i=1,2,\cdots ,N\right\}$, is obtained by averaging non-overlapping window data points of increasing scales. N is the length of time series X, and the time series $y_{j}^{\left(\tau \right)}$ obtained by coarse-graining is expressed by Equation (7):(7)$$y_{j}^{\left(\tau \right)}=\frac{1}{\tau }\sum_{i=(j-1)\tau +1}^{j\tau }x_{i}$$

In Equation (7), τ represents the scale factor, whose range is $1\leq \tau \leq \frac{N}{\tau }$. When τ=1, the coarse-grained sequence is equal to the original sequence.

(2) At various scales, the permutation entropy of coarse-grained sequences is calculated. First, an *m*-dimensional reconstruction of the correlation space is performed on the coarse-grained sequence $\left\{y_{j},j=1,2,\cdots ,m\right\}$:(8)$$Y_{j}=\left\{y_{j},y(j+\tau ),\cdots ,y(j+(m-1)\tau )\right\}$$
where *m* is the embedded dimension and τ represents the time delay. For each *j*, the *m*-dimensional real sequence in ascending order is $y(j+(j_{1}-1))\tau \;\leq \;y(j+(j_{2}-1))\tau \;\leq \;$$y(j+(j_{m}-1))\tau$.

For any $Y_{j}$, a set of symbol sequences, $S(r)=(j_{1},j_{2},\cdots ,j_{m})$, can be obtained; for which r=1,2,⋯,k and k≤m!; wherein m! is the maximum number of permutations of different symbol sequences in the embedding dimension; and *m*. $P_{r}=(r=1,2,\cdots ,k)$ is defined as the probability of occurrence of any symbol sequence S(r). The permutation entropy of any symbol sequence is defined in the form of Shannon entropy:(9)$$H_{p}(m)=-\sum_{r=1}^{k}P_{r}\ln P_{r}$$
where $\sum_{r=1}^{k}P_{r}=1$ and the maximum value of $H_{p}(m)$ is ln(m!). When all the symbol sequences have the same probability, $H_{p}(m)$ is normalized according to Equation (10):(10)$$0\leq H_{p}=H_{p}(m)/\ln(m!)\leq 1$$

### 2.3. Particle Swarm Optimization–Least Squares Support Vector Machine

An LSSVM is an extension of an SVM. It not only reduces computational complexity significantly, but also has strong anti-interference capability and excellent robustness. LSSVMs are widely employed for mechanical equipment fault diagnosis [23]. Assuming the training data set is $x_{i},y_{i}$, where $x_{i}$ is n-dimensional training data, $y_{i}$ is the model input, and l is the number of training samples, the LSSVM can be expressed by Equation (11):(11)$$y(x)=\omega^{T}\theta +b$$

In Equation (11), θ is the feature map, which converts the complex nonlinear relationship between the output, y, and the input, x, into a linear relationship between y and θ. b represents the bias vector, and ω is the weight vector. Equations (12) and (13) can be obtained through optimization:(12)$$\min_{\omega ,b,e}J(\omega ,e)=\frac{1}{2}{\left\|\omega \right\|}^{2}+\frac{1}{2}\gamma \sum_{i=1}^{n}e_{i}^{2}$$
(13)$$s.t.y_{i}=(\omega ,\phi (x_{i}))+b+e_{i}$$

In Equations (12) and (13), γ represents the regularization parameter, $e_{i}$ denotes the error term, and ϕ(x) reflects the nonlinear mapping function from the original space to the multi-bit feature space. The Lagrange function is defined in Equation (14):(14)$$L=\frac{1}{2}{\left\|\omega \right\|}^{2}+\frac{1}{2}\gamma \sum_{i=1}^{n}e_{i}^{2}-\sum_{i=1}^{N}\alpha_{i}\left\{y_{i}\left[\omega^{T}\phi (x_{i}+b)-1+\varepsilon_{i}\right]\right\}$$

The partial derivatives of $\omega ,\;b,\;\alpha_{i},\;\;\mathrm{and}\;\varepsilon_{i}$ are set to 0 and substituted into Equation (11) to obtain the matrix equation:(15)$$\left[\begin{array}{cc} 0 & -y^{T}\\ y & zz^{T}+\gamma^{-1}I \end{array}\right]\left[\begin{array}{l} b\\ a \end{array}\right]=\left[\begin{array}{l} 0\\ I_{n} \end{array}\right]$$

The optimal classification function is defined in Equation (16):(16)$$f(x)=\mathrm{sgn}\left[\sum_{i=1}^{N}\alpha_{i}y_{i}K(x,x_{i})+b\right]$$
where N indicates the number of SVMs, b signifies the classification threshold, and sgn represents the sign function.

A population-based stochastic intelligent algorithm with decent robustness, swift convergence, and powerful global searching capabilities is particle swarm optimization (PSO) [24]. PSO is based on the principle that the optimal solution can be observed through cooperation and information sharing among group members. Through using a PSO algorithm, the LSSVM was optimized to avoid blindness in parameter selection. In accordance with Equation (15), the PSO revises the individual extremum and the group extremum to determine the search direction for each particle. As the iteration time increases, new particle swarms are generated to approach the target value in a continuous manner.
(17)$$\left\{ \begin{array}{c} V_{id}^{k+1}=\omega V_{id}^{k}+c_{1}r_{1}(P_{id}^{k}-X_{id}^{k})+c_{2}r_{2}(P_{gd}^{k}-X_{id}^{k})\\ X_{id}^{k+1}=X_{id}^{k}+V_{id}^{k+1} \end{array}\right.$$

In Equation (17), k represents the number of iterations; $X_{id}$ and $V_{id}$ are the position and position increment vector of the particle in the *k*^th^ iteration, respectively; $c_{1}$ and $c_{2}$ reflect acceleration coefficients; $r_{1}$ and *r*_2_ are two independent random numbers within [0, 1]; and $P_{id}$ and $P_{gd}$ are the local optimum and global optimum in the *k*^th^ iteration, respectively.

A flowchart for the PSO–LSSVM is shown in Figure 2, and the specific steps are described next.

Step (a): Parameters, including the population size, the number of iterations, and the initial position of the population, are initialized.

Step (b): The fitness of the particles is calculated. Using the LSSVM training sample corresponding to each particle, the current particle’s fitness is determined by its recognition error.

Step (c): The fitness of the current particle is compared with the optimal fitness of the entire swarm, followed by a search for the local and global optima. Likewise, the optimal fitness of the present particle is compared to that of the swarm.

Step (d): The positions of the particle and particle swarm are updated based on the fitness of the particle and particle swarm, and the velocity of each particle is also updated accordingly.

Step (e): It is determined whether the iteration termination condition is settled; if not, the model returns to step (b) and continues.

## 3. Fault Diagnosis Model for Internal Gear Pumps

Figure 3b depicts the CAE–VMD–MPE–PSO–LSSVM fault diagnosis model for internal gear pumps, and the individual steps are described below.

(a) An internal gear pump is subjected to accelerated life testing, the vibration acceleration signal is collected for its entire life cycle, and the operating periods are classified premised on the signal features.

(b) A CAE is utilized to preprocess signals in different operating periods, suppress background noise, and enhance the signal’s fault features. Figure 3a and Table 1 both illustrate the CAE architecture.

(c) A VMD–MPE method is utilized to reconstruct the fault features of the signal, decompose the signal into multiple IMF components using VMD, and calculate the MPE of each IMF to generate a feature set.

(d) The feature set is divided into a training set and a test set, the test set is trained by a PSO–LSSVM, and the fitness curve is utilized to select the appropriate parameters. Thereafter, pattern recognition is carried out for the test set using the trained model, and the results are evaluated against other available fault diagnosis techniques.

## 4. Experiments and Results

### 4.1. Accelerated Life Testing

The subject of this study is a particular type of linear conjugate internal gear pump. The pump was an innovative three-port internal gear pump, and marine hydraulic systems utilize this type of pump due to its low flow pulsation and four-quadrant operation capability. Figure 4 depicts the internal structure of the internal gear pump, and Table 2 lists the technical parameters.

Overspeed and overload shock testing and performance testing comprised the accelerated life testing of the internal gear pump. In the first stage, the internal gear pump was subjected to overspeed and overload to simulate the internal gear pump’s deterioration over the course of its entire life cycle. The system oil temperature was set to 80 °C, the rotational speed was 3000 rpm, and the frequency of the shock pressure was 40 times/min at 20 MPa. During the 1.5-s-duration of a single shock, the loading and unloading durations were 1.0 s and 0.5 s, respectively. ACFTD air filter fine test dust (0~20 μm) was added to the oil to increase its cleanliness to NAS11 standards.

During the performance testing phase, a 12-channel B&K module was used to acquire the vibration acceleration signal on the shaft side of the internal gear pump. The test speed was 1800 rpm, the test pressure was 8 MPa, the sampling frequency was 32,768 Hz, and the sampling duration was 30 s. The internal gear pump was considered to have failed the accelerated life test when its volumetric efficiency was less than 82.5% at 1800 rpm, 8 MPa, and 80 °C oil temperature.

Figure 5 depicts the process of accelerated life testing and gear ring wear. As it transitioned from a normal state to a complete failure, the internal gear pump experienced a total of 780,000 shocks. The test lasted for 53 days, and the total shock duration was 325 h. Initially, the performance was evaluated prior to accelerated life testing, and then it was assessed every 20,000 shocks during the accelerated life testing procedure. Throughout the duration of the test cycle, 40 performance tests were conducted.

The life cycle of the pump was divided into four operating periods: an initial run-in period, a stable operating period, an early failure period, and a terminal failure period, based on the authors’ prior research regarding this type of internal gear pump and data from accelerated life testing. Due to space constraints, these eras cannot be described in detail here. Table 3 lists the performance tests and sample numbers for the four time periods. There is a significant difference between the number of data points for the pump’s different operating periods. The number of samples during the final failure period is particularly low, posing difficulties in diagnosing faults in the internal gear pump.

### 4.2. CAE-Based Signal Preprocessing

The operating information for the internal gear pump primarily includes the rotational frequency of the shaft, $f_{c}$, the meshing frequency, $f_{b}$, of the ring gear, and their multiplication frequencies. Figure 6 depicts the time-domain and frequency-domain signals for the internal gear pump during its four operating phases. In general, the signal-to-noise ratio of the raw signal in the four periods is low, and the signal is subject to significant mechanical vibration and environmental noise influences, causing $f_{c}$, $f_{b}$, and the multiplication frequencies to be obscured by the noises. Therefore, the signal cannot accurately reflect the operating data of the pump. Specifically, the signals during the initial run-in period and the early failure period have a large number of line spectrum clusters that are unrelated to the operating state of the internal gear pump, and their amplitudes are very high. If the signal features are extracted directly to identify the different operating periods of an internal gear pump, the accuracy will be low.

Figure 7 depicts the reconstructed signal, which was generated by processing the raw signals for the four periods using a CAE. The figure demonstrates how the rotation frequency, $f_{c}$, the meshing frequency, $f_{b}$, and multiplication frequencies in the CAE-reconstructed signal are enhanced in comparison to the original signal, while the vibration and ambient noise are effectively suppressed. Consequently, the operational specifics of the internal gear pump are highlighted. The signal-to-noise ratio of the reconstructed signal is 6.4–8.7 dB higher than the signal-to-noise ratio of the original signal, indicating that the CAE has a positive effect and establishes a solid foundation for subsequent feature extraction and pattern recognition.

### 4.3. VMD–MPE-Based Signal Feature Extraction

After CAE-based signal reconstruction, signal features were extracted with the VMD–MPE method to reflect the operating information of the internal gear pump. The number of VMD decomposition layers, K, significantly influences the decomposition results. If K is small, important information may be lost, and if K is large, there could be an over-decomposition problem [24]. Observing the center frequency of each mode revealed that when K=7, the IMF6 and IMF7 center frequencies are close, resulting in over-decomposition. Hence, K=6 was employed in this study.

The signals for the four periods of the internal gear pump were subjected to VMD decomposition. Figure 8 depicts the frequency-domain waveform during the early failure period. Without modal aliasing, the center frequencies of the different modes are dispersed. Thus, the decomposition of signals with varying center frequencies was accomplished.

Before extracting the MPE value of each signal IMF, it is necessary to set the time delay, τ, the scale factor, s, and the embedding dimension, m. According to Wang et al. [25], τ has little effect on the time series analysis results. If m is too small or too large, it may hinder the algorithm’s ability to detect signal mutations or small time series changes. Generally, s should be greater than 10. After a thorough analysis of the effects of the three parameters on the MPE, we set τ=1, s=20, and m=6. Then, an MPE analysis was performed on each IMF for each vibration signal, and each sample formed a vector of 120-dimensional features.

Figure 9 displays the MPE values for the four operating periods of the internal gear pump in six IMFs. Figure 9 illustrates two conclusions. First, when s≥10, the recognition results for the four periods’ signals are promising. Second, with the exception of the aliasing of the MPE values for the various periods in IMF5, the signal differences between the four periods are readily apparent in the remaining five IMFs. The outcome demonstrates the efficiency of the VMD–MPE method for feature extraction.

### 4.4. PSO–LSSVM-Based Pattern Recognition

The size of the data set in the original vibration acceleration signal is 32,768. After feature extraction by VMD-MPE, the size of its feature set is 120, the number of sample sets of internal gear pump in different running states is 1200, and the dimension of feature set of internal gear pump in four states is 120 × 1200. The number of feature sets in the initial run-in period is 270, the number of feature sets in the stable operation period is 660, the number of feature sets in the early failure period is 240, and the number of feature sets in the late failure period is 30. After the feature sets of internal gear pump in different states were randomly scrambled, 80% of the feature sets of internal gear pump in each state were taken as the training set, and the remaining 20% were taken as the test set. Therefore, the number of training sets is 960 and the number of test sets is 240.

The training set was input into the PSO-LSSVM model for training and classification. The training utilized PSO–LSSVM. During the training process, the initial population number was set to 20, the i26teration time was 200, and the acceleration factors, $c_{1}$ and $c_{2}$, were 1.5 and 1.8, respectively [23,26]. The fitness curve for the training process is depicted in Figure 10. The optimal fitness of the training samples was 100%, the kernel function parameter, c, was 0.1 after PSO optimization, and the penalty factor, *g,* was 663.833.

Figure 11 depicts the results of pattern recognition of the fault state in the test sets using the trained PSO–LSSVM model. The diagnosis rates for the internal gear pump’s four operating periods are 98.1%, 100%, 97.9%, and 100%, respectively. There were only two incorrect identifications among the 240 data sets: one data set from the initial running-in period was misclassified as an early failure period. In the early failure period, one data set was classified as the initial running-in period. The remaining classifications were all appropriate. Overall, the average diagnosis rate for the internal gear pump’s four operational periods was 99.2%.

To further validate the efficacy of the CAE–VMD–MPE–PSO–LSSVM method, it was compared to three other methods: VMD–MPE–PSO–LSSVM, CAE–MPE–PSO–LSSVM, and CAE–VMD–MPE–LSSVM. The results are shown in Figure 12. Each of the three methods has a diagnosis rate that is lower than the CAE-VMD-MPE-PSO-LSSVM method: 93.8%, 96.3%, and 96.3%, respectively. In addition, the diagnosis rate is significantly decreased when the raw signal is not denoised by a CAE, proving that CAE preprocessing of a signal with a low signal-to-noise ratio is effective for enhancing the fault diagnosis accuracy. However, CAE, as a deep learning model, is used for the preprocessing of vibration signals, which have a large amount of calculation and high model complexity. VMD is also used for signal decomposition, which has a large amount of calculation, resulting in the CAE-VMD-MPE-PSO-LSSVM method for internal gear pump fault diagnosis. There are also problems such as low computational efficiency and high model complexity, and it is difficult to achieve real-time monitoring of the health status of internal gear pumps. This issue can also be the focus of further research.

## 5. Conclusions

In light of the issue that the signals of internal gear pumps can be easily obscured by mechanical vibrations and environmental noise, as well as the fact that the number of fault samples is small, initially, an internal gear pump was subjected to accelerated life testing and signals were collected during different operating periods. Secondly, the raw signal was preprocessed using CAE, which accomplished the suppression of noise and enhancement of features representing the pump’s operating state. The preprocessed signal was then decomposed into multiple IMFs utilizing VMD, and the MPE value of each IMF was extracted individually to establish a feature set. Finally, a PSO–LSSVM was utilized for pattern recognition of the internal gear pump signals to achieve precise fault diagnosis. Three primary conclusions were reached after comparing the CAE–VMD–MPE–PSO–LSSVM method to other fault diagnosis techniques:(1)The CAE–VMD–MPE–PSO–LSSVM fault diagnosis model accurately determined the operating state of the internal gear pump; consequently, accurate fault diagnosis was accomplished. A comparative analysis revealed that the proposed method for diagnosing faults in internal gear pumps is more effective and accurate than other methods.(2)Utilizing a CAE to preprocess the raw signal of an internal gear pump in an environment with complex noise exhibits a positive effect. Effectively suppressing background noise and enhancing operating state features lays a solid foundation for subsequent feature extraction and pattern recognition.(3)A comparison of the MPE values of the internal gear pump during different operating periods demonstrates that the MPE method is robust and anti-interference is strong, and the signal can be analyzed at various time scales. Therefore, it is possible to accurately characterize the operating state of the internal gear pump.

## Figures and Tables

**Figure 1 sensors-22-09841-f001:**
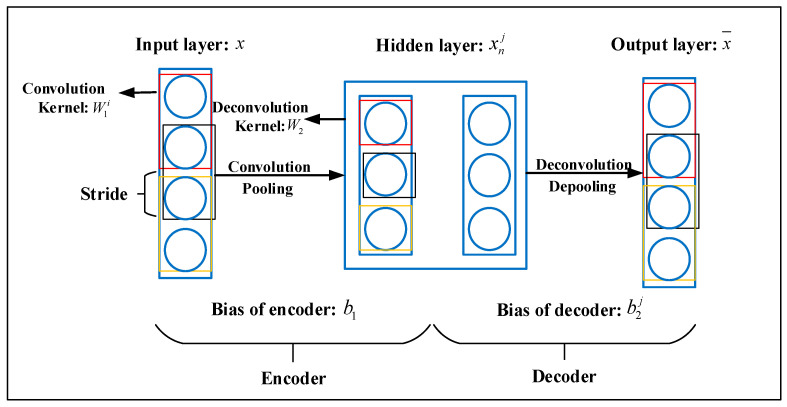
The network structure of convolutional auto-encoder.

**Figure 2 sensors-22-09841-f002:**
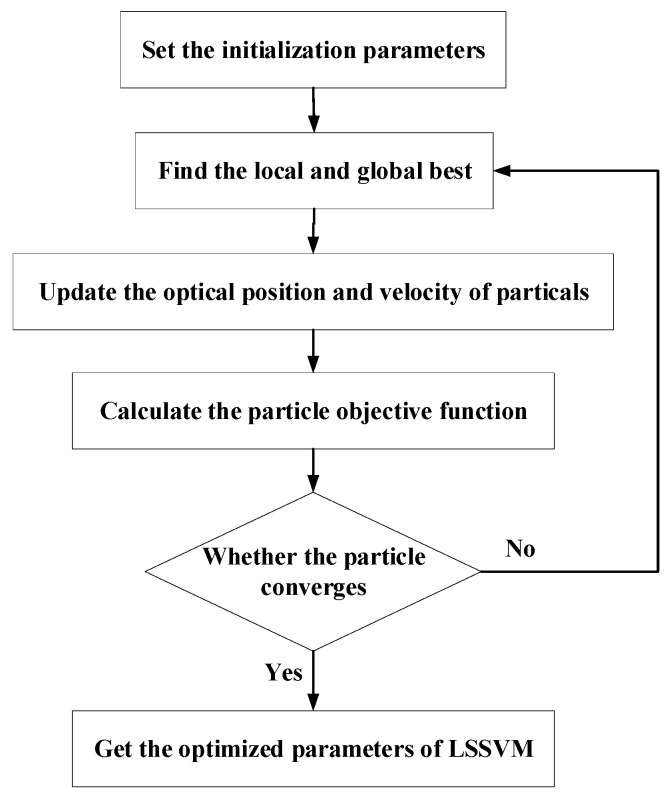
Flowchart of PSO-LSSVM.

**Figure 3 sensors-22-09841-f003:**
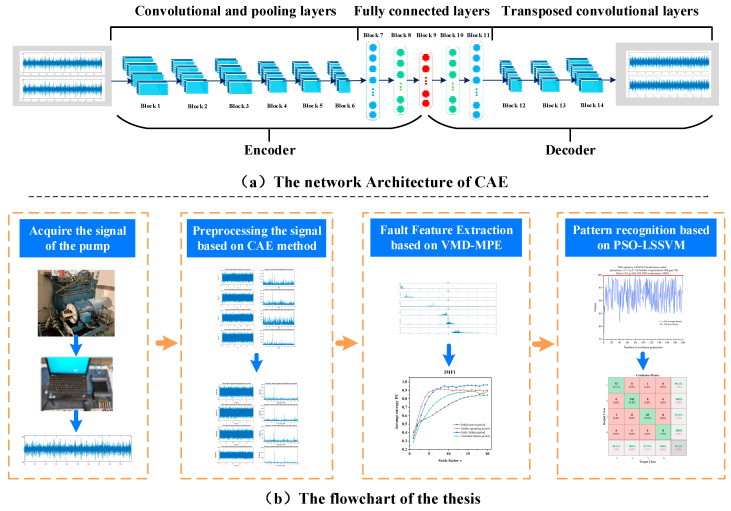
CAE architecture and flow chart.

**Figure 4 sensors-22-09841-f004:**
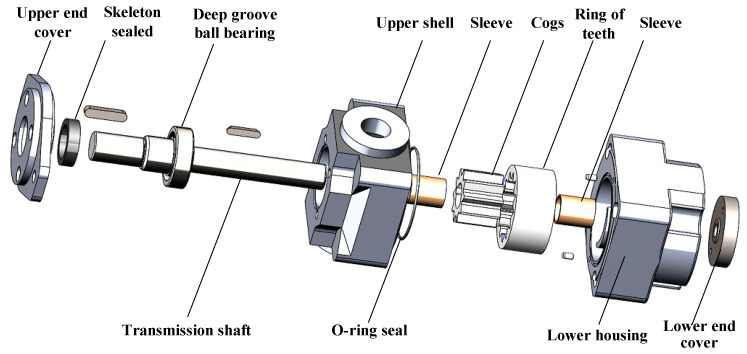
Structure of the internal gear pump.

**Figure 5 sensors-22-09841-f005:**
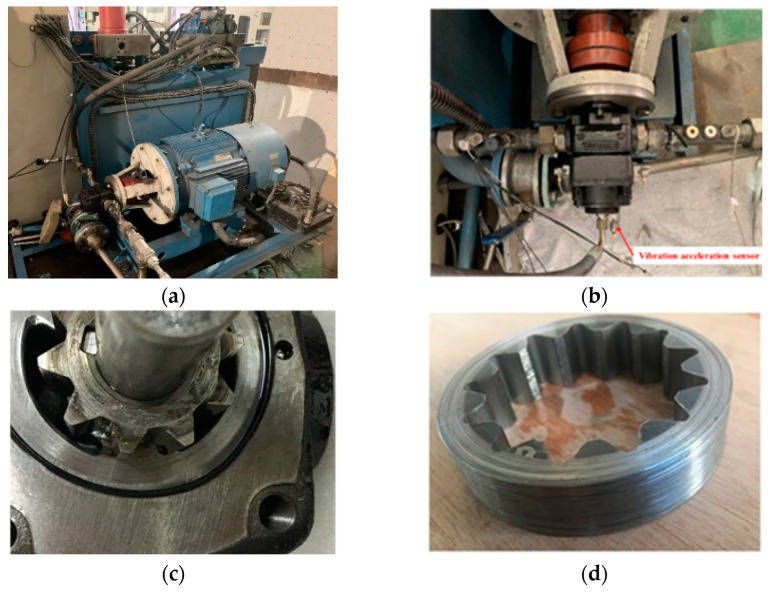
Accelerated life testing process. (**a**) Test setup. (**b**) sensor placement. (**c**) gear wear. (**d**) gear ring wear.

**Figure 6 sensors-22-09841-f006:**
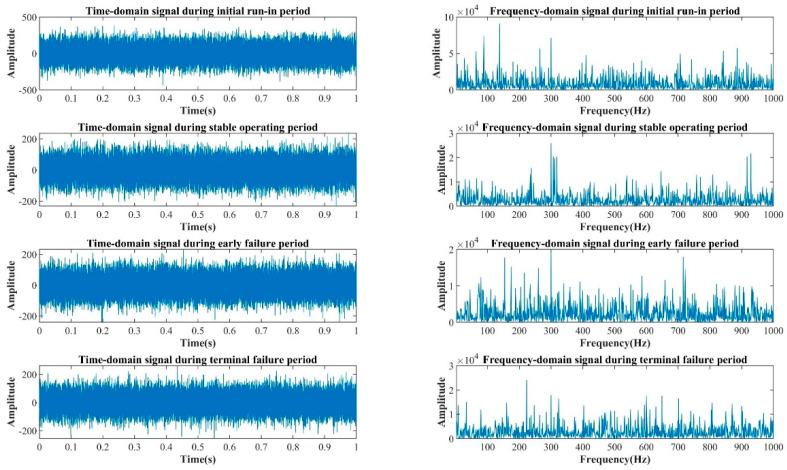
Raw signals of the internal gear pump in four operating periods.

**Figure 7 sensors-22-09841-f007:**
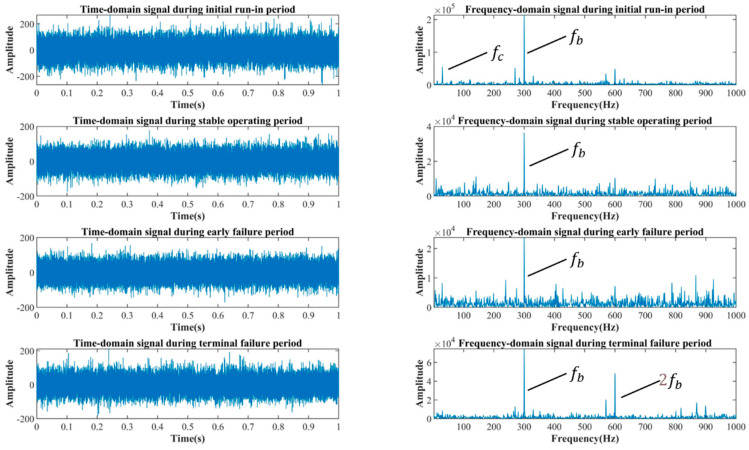
Internal gear pump signals in the four operating periods after CAE reconstruction.

**Figure 8 sensors-22-09841-f008:**
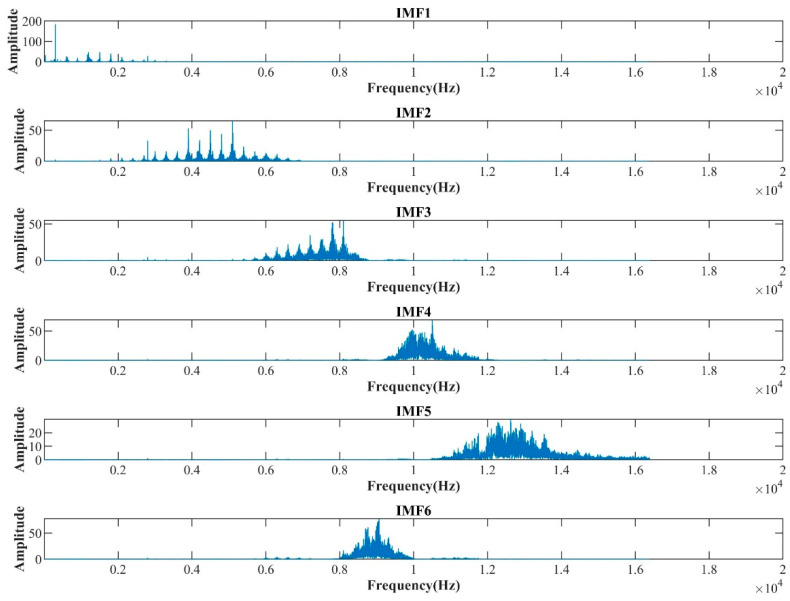
Frequency-domain signal in the early failure period after VMD decomposition.

**Figure 9 sensors-22-09841-f009:**
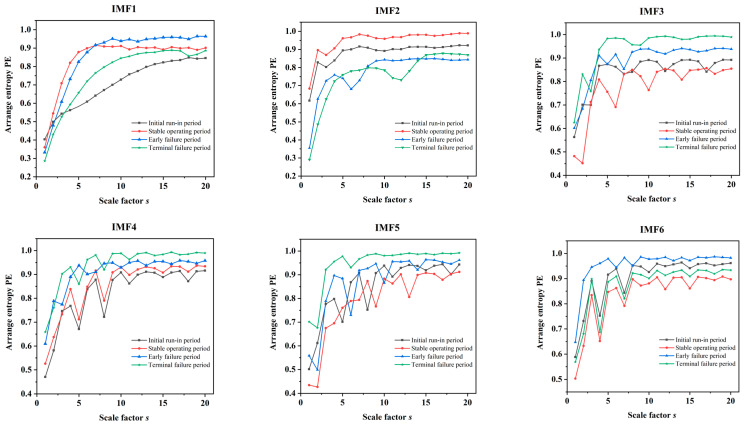
MPE values of the internal gear pump in the four operating periods under different modes.

**Figure 10 sensors-22-09841-f010:**
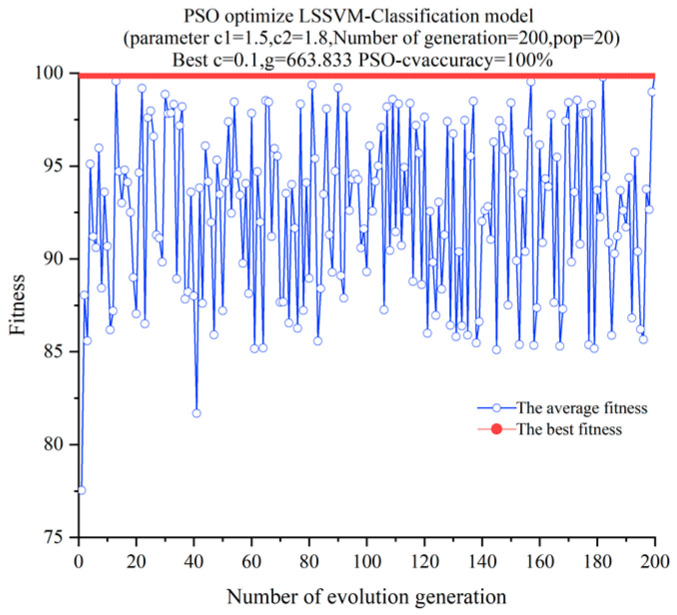
PSO–LSSVM fitness curve.

**Figure 11 sensors-22-09841-f011:**
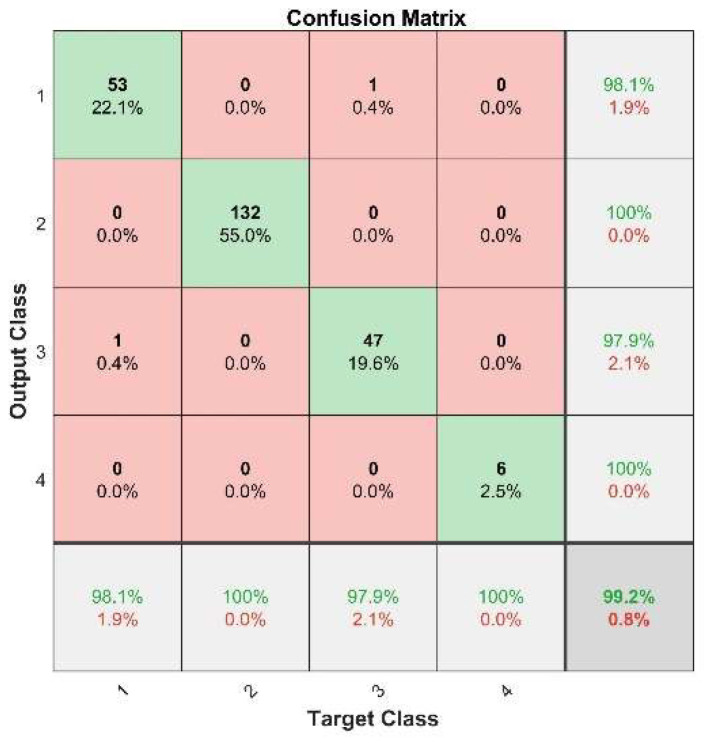
Confusion matrix of the results of the CAE–VMD–MPE–PSO–LSSVM method.

**Figure 12 sensors-22-09841-f012:**
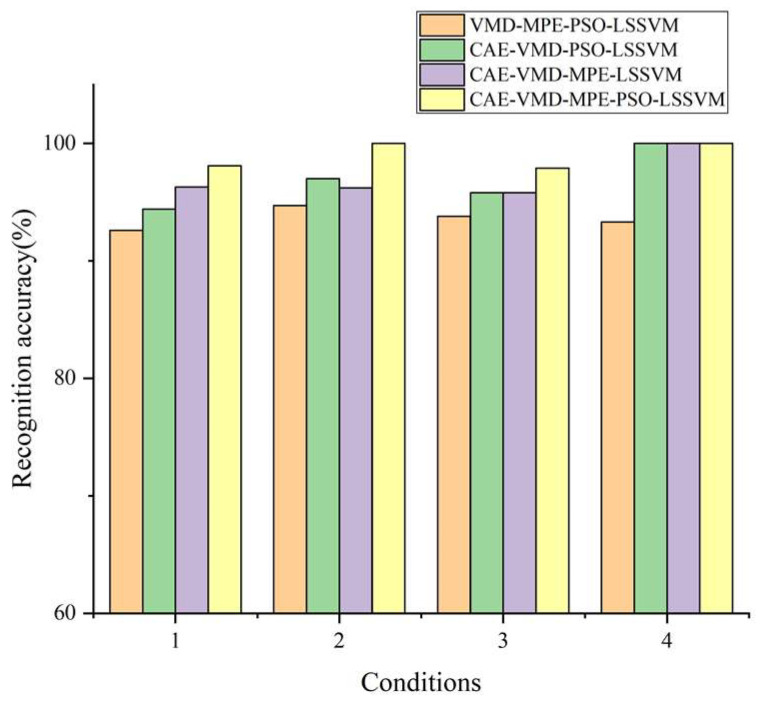
Accuracy rates for the four fault diagnosis methods.

**Table 1 sensors-22-09841-t001:** Detailed configuration parameters of CAE.

Type	Layer	Receptive Field Size/Stride/Number of Channels	Activation Function
Block1	Convolution	(3, 3)/1/32	ReLU
Block2	Pooling	(2, 2)/2/32	ReLU
Block3	Convolution	(3, 3)/1/64	ReLU
Block4	Pooling	(2, 2)/2/64	ReLU
Block5	Convolution	(3, 3)/1/128	ReLU
Block6	Pooling	(2, 2)/2/128	ReLU
Block7/Block11	FC	1/*/1024	ReLU
Block8/Block10	FC	1/*/512	ReLU
Block9	FC	1/*/128	Sigmoid
Block12	Deconvolution	(3, 3)/1/64	ReLU
Block13	Deconvolution	(3, 3)/1/32	ReLU
Block14	Deconvolution	(3, 3)/1/1	Sigmoid

**Table 2 sensors-22-09841-t002:** Primary technical parameters of the internal gear pump.

Displacement	Maximum Speed	Rated Pressure	Maximum Torque
20 mL/r	3000 rpm	16 MPa	50 Nm

**Table 3 sensors-22-09841-t003:** Operating periods of the internal gear pump and numbers of samples.

Life Cycle Classification	Performance Tests	Number of Datasets × Dataset Size
Initial run-in period	[1–9]	270 × 32,768
Stable operating period	[10–31]	660 × 32,768
Early failure period	[32–39]	240 × 32,768
Terminal failure period	40	30 × 32,768

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
