# Peer review of "Research on the Fault Diagnosis Method of an Internal Gear Pump Based on a Convolutional Auto-Encoder and PSO-LSSVM"

_sensors, 2022, doi:10.3390/s22249841_

Round 1

Reviewer 1 Report

The paper focuses on a new fault Diagnosis method of Internal Gear Pump Based on Convolutional Auto-Encoder and PSO-LSSVM. Experimental investigation was carried out to test the proposed method.

The topic is interesting and original especially for internal gear pump diagnosis. The paper shows the efficiency of PSO-LSSVM method to diagnose gear pump defects with a new approach. The references are suitable and appropriate. Some figures should be enlarged for clarity. Some aspects should be further discussed by the authors:

- Why the VMD–MPE is selected to extract signal features

- Is the proposed method efficient for non stationary regimes

Author Response

RESPONSES TO REVIEWER1

1- Why the VMD–MPE is selected to extract signal features

Response: Thank you very much for your suggestion.

VMD has the advantages of better complex data decomposition accuracy and better anti-noise interference, and can realize signal decomposition. MPE has the characteristics of good robustness, strong anti-interference and simple calculation of permutation entropy for complex time series, and can realize the analysis of signals at different time scales and obtain more comprehensive information.

The feature extraction of the signal by VMD-MPE converts the original data set matrix from 1200 × 32768 to 1200 × 120. The selection of VMD-MPE for signal feature extraction can accurately extract the characteristics that can reflect the running state of the pump, and can also reduce the dimension of the feature set, so as to reduce the complexity of the model for subsequent fault identification and reduce the amount of calculation.

2- Is the proposed method efficient for non stationary regimes

Response: At present, the research we have carried out is mainly aimed at the fault diagnosis of the internal gear pump under steady working conditions and the fault diagnosis of the internal gear pump under non-stationary conditions is still challenging. We will also study the fault diagnosis of internal gear pump under variable working conditions in the next step.

In addition, we have made a lot of modifications and polishes to the language of the manuscript, and also supplemented some details.

Reviewer 2 Report

The manuscript is well structured and well argued. However, several rectifications and modifications are required to ensure that its quality stands up to this reputed journal.  

1.       The authors have proposed an improved fault diagnosis method based on convolutional auto-encoder and PSO-LSSVM for accurately diagnosing faults in internal gear pumps.

2.       The English language must be improved. There are several grammatical errors as one goes through the manuscript that requires rectification. Most of the sentences convey no proper meaning and could be off-putting to the readers and practitioners.

3.       The first section introduces a basic outlook on related works of numerous signal pre-processing methods and deep learning techniques to extract signal features from raw data and a brief literature review of various methodologies used for diagnosing faults in internal gear pumps is presented.

4.       The resolutions of all the figures of manuscript are required to be improved.

5.       Authors have to clearly mention the details of how the training set and test set are divided of data used?

6.       What are the criteria for selecting parameters of Particle Swarm Optimizer?

7.       Suggested to compare the performance of proposed method with recently reported methods for diagnosing faults in internal gear pumps with respect to the accuracy and diagnosis rates. Preferably include table.

8.         To keep things fair, a brief discussion of the demerits of the proposed analysis should be provided.

9.       The manuscript has the potential to be improved and requires minor rectifications. With that being said, I wish the authors all the best in their endeavor to improve the quality of the manuscript.

Author Response

RESPONSES TO REVIEWER2

  1. The English language must be improved. There are several grammatical errors as one goes through the manuscript that requires rectification. Most of the sentences convey no proper meaning and could be off-putting to the readers and practitioners.

Response: Thank you very much for your suggestion, we have made a large number of comprehensive revisions to the manuscript, focusing on grammatical errors and inappropriate expressions, in order to achieve accurate expression of the manuscript. The revised content can be seen in the latest version of the uploaded manuscript.

  1. The resolutions of all the figures of manuscript are required to be improved.

Response: We have changed most of figures of manuscript in the manuscript to make them more high-resolution, including figure 6 to 12. The first 5 pictures are mainly the pictures made by viso and the test results, It may be due to the software or camera configuration. The revised figures can be seen in the latest version of the uploaded manuscript.

  1. Authors have to clearly mention the details of how the training set and test set are divided of data used?

Response: We have added the details of the division of training set and test set in Section 4.4 of the manuscript, which is as follows:

The size of the data set in the original vibration acceleration signal is 32768. After feature extraction by VMD-MPE, the size of its feature set is 120, the number of sample sets of internal gear pump in different running states is 1200, and the dimension of feature set of internal gear pump in four states is 120×1200. The number of feature sets in the initial run-in period is 270, the number of feature sets in the stable operation period is 660, the number of feature sets in the early failure period is 240 and the number of feature sets in the late failure period is 30. After the feature sets of internal gear pump in different states were randomly scrambled, 80% of the feature sets of internal gear pump in each state were taken as the training set, and the remaining 20% were taken as the test set. Therefore, the number of training sets is 960 and the number of test sets is 240.

  1. What are the criteria for selecting parameters of Particle Swarm Optimizer?

Response: We mainly based on the references related to the PSO-LSSVM algorithm, especially the journal papers related to the fault diagnosis of rotating machinery, to determine the parameters. as follows: the initial population number was set to 20, the iteration time was 200, and the acceleration factors,  and , were 1.5 and 1.8. The correctness of parameter selection is verified by experimental data. Relevant references have also been added to the manuscript. Here it is:

[23] Chen W, Li J , Wang Q , et al. Fault Feature Extraction and Diagnosis of Rolling Bearings Based on Wavelet Thresholding Denoising with CEEMDAN Energy Entropy and PSO-LSSVM, Measurement, 2020.

[26] Hongbo Xu, Guohua Chen, An intelligent fault identification method of rolling bearings based on LSSVM optimized by improved PSO, Mechanical Systems and Signal Processing, 2013, 35 167-175.

  1. Suggested to compare the performance of proposed method with recently reported methods for diagnosing faults in internal gear pumps with respect to the accuracy and diagnosis rates. Preferably include table.

Response: Your suggestion is constructive, and we would like to compare proposed method with recently reported methods, but there are two main reasons that limit us.

First, because our research object is a special linear conjugate internal gear pump, which is mainly used in military ships and is not widely used in the industrial field, there is less research on the fault diagnosis of this type of pump, which is not convenient for direct comparison;

Second, due to the lack of a unified internal gear pump operating state data set in the field of mechanical fault diagnosis, there are some differences in the vibration characteristics of pumps in different hydraulic systems, which restrict our ideas.

  1. To keep things fair, a brief discussion of the demerits of the proposed analysis should be provided.

Response: At the end of section 4.4 of the manuscript, we add a description of the shortcomings of the proposed method, mainly due to the large amount of calculation and the high complexity of the model. The description part in the manuscript is as follows.

However, CAE, as a deep learning model, is used for the preprocessing of vibration signals, which has a large amount of calculation and high model complexity. VMD is also used for signal decomposition, which has a large amount of calculation, resulting in CAE-VMD-MPE-PSO-LSSVM method for internal gear pump fault diagnosis. There are also problems such as low computational efficiency and high model complexity, and it is difficult to achieve real-time monitoring of the health status of internal gear pumps. This issue can also be the focus of further research.
